# The Missing Invariance Principle Found –
# the Reciprocal Twin of Invariant Risk Minimization

**Dongsung Huh**[*]
MIT-IBM Watson AI Lab
Cambridge, MA 02142
huh@ibm.com

**Avinash Baidya**[*]
Department of Physics and Astronomy
University of California
Davis, CA 95616
aavinash@ucdavis.edu

## Abstract

Machine learning models often generalize poorly to out-of-distribution (OOD) data as a result of relying on features that are spuriously correlated with the label during training. Recently, the technique of Invariant Risk Minimization (IRM) was proposed to learn predictors that only use invariant features by conserving the feature-conditioned label expectation $\mathbb{E}_e[y|f(x)]$ across environments. However, more recent studies have demonstrated that IRM-v1, a practical version of IRM, can fail in various settings. Here, we identify a fundamental design flaw of IRM formulation that causes the failure. We then introduce a complementary notion of invariance, MRI, based on conserving the label-conditioned feature expectation $\mathbb{E}_e[f(x)|y]$, which is free of this flaw. Further, we introduce a simplified, practical version of the MRI formulation called MRI-v1. We prove that for general linear problems, MRI-v1 guarantees invariant predictors given sufficient number of environments. We also empirically demonstrate that MRI-v1 strongly out-performs IRM-v1 and consistently achieves near-optimal OOD generalization in image-based nonlinear problems.

## 1 Introduction

Deep learning models have shown tremendous success over the past decade. These models show great generalization properties when tested on the same distribution as the training dataset (in-distribution generalization). However, these models often show catastrophic failure when tested on out-of-distribution dataset, revealing that they learned features that are spuriously correlated to the label in the given training domains but do not generalize to the testing domains. For example, deep networks trained on pictures of cow with only grassy backgrounds in the training domain will use the background color as the predictive feature which is easier to learn and generalize poorly to pictures of cow with a dessert background.

Recently, there has been a growing interest in developing models that generalize well across multiple domains. In particular, there has been a recent body of works that focus on developing algorithms that attempt to learn invariant predictors (Arjovsky et al., 2019; Ahuja et al., 2021; Peters et al., 2016; Rojas-Carulla et al., 2018; Heinze-Deml et al., 2018). Invariant Risk Minimization (IRM) and its practical version, IRM-v1, have garnered significant attention as one of the initial methods that are compatible with deep learning techniques. However, several follow-up studies have empirically demonstrated that IRM-v1 is unreliable at learning invariant representations (Kamath et al., 2021; Rosenfeld et al., 2020; Gulrajani and Lopez-Paz, 2020; Ahuja et al., 2020, 2021). Here, we identify a fundamental flaw of IRM formulation that causes this limitation and propose a new method that is free of this flaw.

---

[*]Equal contribution

36th Conference on Neural Information Processing Systems (NeurIPS 2022).

**Related Works** There has been considerable work in the field of learning invariant representations. They vary from learning domain-invariant feature representations conserving $P(f(x))$ using kernel methods (Muandet et al., 2013; Ghifary et al., 2016; Hu et al., 2020), variational autoencoder (Ilse et al., 2020), and adversarial networks Ganin et al. (2016); Long et al. (2018); Akuzawa et al. (2019); Albuquerque et al. (2019) to learning invariant class-conditional features $P(f(x)|y)$ (Gong et al., 2016; Li et al., 2018b) in the context of domain adaptation, which assumes access to the test distribution for adaptation. There is also a large body of work to learn invariant representations in the field of domain generalization that doesn't assume access to test distributions. This includes imposing invariance of $\mathbb{E}_e[y|f(x)]$ (Arjovsky et al., 2019) with information bottleneck constraint (Ahuja et al. (2021)), imposing object-invariant condition (Mahajan et al. (2021)), using domain inference (Creager et al., 2021), model calibration (Wald et al., 2021), and others (Krueger et al., 2021; Li et al., 2018a; Shankar et al., 2018).

**Our Contributions** We introduce a variational formulation of IRM, and show that it can be modified to yield a new complementary notion of invariance, called MRI. We show that IRM has a fundamental flaw due to the indirect way of imposing invariance which leads to the failure of IRM-v1. In constrast, MRI is shown to be free of this flaw. We prove that MRI-v1 can guarantee invariant predictors in general linear problem settings given sufficient environments. We also show empirical demonstrations that MRI strongly out-performs IRM and consistently achieves near-optimal OOD generalization in nonlinear image-based problems.[2]

## 2 Problem Formulation

Consider a set of environments $\mathcal{E} = \{e\}$, each of which defines a distribution $P_e(x, y)$ over inputs and labels, from which the dataset of the environment is drawn $\mathcal{D}_e \equiv \{(x_j^e \in \mathbb{R}^d, y_j^e \in \mathbb{R})\}$. The distribution $P_e(x, y)$ is assumed to be generated according to the causal graph in Fig 1 (Rosenfeld et al., 2020), which includes latent features that are invariantly $z_i$ or spuriously $z_s$ correlated with the label, from which the observation $x$ is generated.[3]

The risk of a predictor $f : X \to O$ in environment $e$ is defined as the population average

$$\mathcal{L}_e(f) \equiv \mathbb{E}_{P_e(x,y)}[l(f(x), y)] \tag{1}$$

$$= \mathbb{E}_{P_e(o)}[\mathbb{E}_{P_e(y|o)}[l(o, y)]] \tag{2}$$

$$= \mathbb{E}_{P(y)}[\mathbb{E}_{P_e(o|y)}[l(o, y)]] \tag{3}$$

where $o = f(x)$ is the predictor's output. Here, we consider standard convex loss functions $l : O \times Y \to \mathbb{R}_{\geq 0}$, including the square loss $l_{sq}(o, y) = \frac{1}{2}(o - y)^2$ for regression $(O, Y \subseteq \mathbb{R})$ and the binary-cross-entropy (BCE) loss $l_{log}(o, y) = -(1 + y) \log(\eta(o)) - (1 - y) \log(1 - \eta(o))$ for binary classification $(O \subseteq \mathbb{R}, Y \subseteq [-1, 1])$, where $\eta(o) \equiv 1/(1 + e^{-o})$ is the sigmoid function.

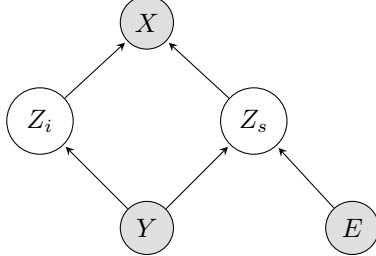

Figure 1: Causal graph depicting the data generating process. Shading indicates the variable is observed.

## 3 IRM vs MRI Invariance

### 3.1 IRM Paradigm

#### 3.1.1 Original formulation

**Definition 3.1.** (Arjovsky et al., 2019) The feature representation $f : X \to Z$ elicits an *IRM invariant* predictor $\psi \circ f : X \to O$ over a set of environments $\mathcal{E}$, if there exists $\psi : Z \to O$ such that $\psi$ is simultaneously optimal for all environments in $\mathcal{E}$, *i.e.*

$$\forall e \in \mathcal{E}, \qquad \psi \in \arg\min_{\bar{\psi}} \mathcal{L}_e(\bar{\psi} \circ f) \tag{4}$$

---

[2]Code available at https://github.com/IBM/MRI.

[3]While Fig 1 only shows the causal direction $Y \to Z_i$, the other direction $Z_i \to Y$ is also consistent with our analysis here as long as $P(y)$ is independent from the environment index $e$.

where $\psi$ is assumed to be unrestricted in the space of all measurable functions.

**Lemma 3.2.** *([Kamath et al., 2021](#)) For standard loss functions[4], Definition 3.1 is equivalent to*

$$\exists \psi, \ \forall e \in \mathcal{E}, \qquad \mathbb{E}_e[y|f(x)] = \sigma(\psi \circ f(x)). \tag{5}$$

*where $\mathbb{E}_e[y|f(x)] \equiv \mathbb{E}_{P_e(y|f(x))}[y]$ is the feature-conditioned label expectation, and $\sigma$ is a monotonic function that depends on the loss function.*

### 3.1.2 Variational formulation

The original formulation above is overly complex due to the composite predictor $\psi \circ f$ and the optimality condition on $\psi$. For further analysis, we introduce the following variational formulation.

**Definition 3.3.** A predictor $f : X \to O$ is *IRM invariant* over a set of environments $\mathcal{E}$, if the risk $\mathcal{L}_e(f)$ remains stationary under arbitrary infinitesimal perturbations on the predictor output $o = f(x)$ for all environments in $\mathcal{E}$, *i.e.*

$$\forall e \in \mathcal{E}, \quad \delta\mathcal{L}_e(f) = \lim_{\epsilon \to 0} \mathbb{E}_{P_e(o,y)}[l(o + \epsilon\delta\psi(o), y) - l(o, y)]/\epsilon$$
$$= \mathbb{E}_{P_e(o)}[\mathbb{E}_{P_e(y|o)}[\partial_o l(o, y)] \cdot \delta\psi(o)] = 0 \tag{6}$$

where $\delta\psi : O \to O$ is an arbitrary perturbation that is unrestricted in the space of all measurable functions, and $\delta\mathcal{L}_e(f)$ denotes the resulting change in risk.

**Lemma 3.4.** *For standard loss functions[4], Definition 3.3 is equivalent to*

$$\forall e \in \mathcal{E}, \qquad \mathbb{E}_e[y|o] = \sigma(o), \tag{7}$$

*which is equivalent to Lemma 3.2 with the composite predictor $\psi \circ f$ replaced by $f$.*

*Proof.* Eq (6) is satisfied if and only if $\mathbb{E}_{P_e(y|o)}[\partial_o l(o, y)] = 0$. For standard loss functions, the loss derivative has the form $\partial_o l(o, y) = -y + \sigma(o)$, which yields $\mathbb{E}_e[\partial_o l(o, y)|o] = -\mathbb{E}_e[y|o] + \sigma(o) = 0$. $\square$

Note that even though Definition 3.3 describes only the first-order condition for the predictor to be simultaneously optimal over all environments, this is indeed the necessary and sufficient condition for optimality, since the loss function $l$ is convex. This yields a simpler formulation of IRM without requiring a composite form for the predictor $\psi \circ f$.

### 3.1.3 Conservation law of IRM

As noted in [Arjovsky et al. (2019)](#); [Kamath et al. (2021)](#), the essence of IRM's invariance is the conservation of the *feature-conditioned label expectation*, i.e.

$$\forall e_1, e_2 \in \mathcal{E}, \qquad \mathbb{E}_{e_1}[y|f(x)] = \mathbb{E}_{e_2}[y|f(x)]. \tag{8}$$

This result can be easily seen from eq (5),(7), since their RHS term $\sigma(o)$ is constant with respect to the environment index $e$.

**Remark** Notice an intriguing discrepancy in the number of constraints: IRM (eq (4),(5),(7)) imposes one constraint per environment, total of $|\mathcal{E}|$ constraints, whereas the conservation law describes $|\mathcal{E}| - 1$ equality relationships that $\mathbb{E}_e[y|o]$ should share the same value across environments. The missing constraint is that IRM additionally requires the shared value of $\mathbb{E}_e[y|o]$ to also be equal to $\sigma(o)$. This discrepancy emphasizes the fact that the equality relationships in eq (8) are not direct constraints imposed to hold between environments, but rather a byproduct — an indirect consequence of separate individual constraints all sharing a common intermediate term, $\sigma(o)$. Furthermore, it shows that the invariance guarantee of IRM singularly depends on the constancy of this shared term across environments, which proves to be a single point of failure for IRM.

---

[4] Square loss and BCE loss are considered: $\sigma(o) = o$ for square loss, and $\sigma(o) = \tanh(o/2)$ for BCE loss.

### 3.1.4 IRM-v1

Due to the impracticality of considering the unrestricted function space of $\psi$, Arjovsky et al. (2019) suggested restricting $\psi$ to the space of linear functions. In the variational formulation, this corresponds to restricting the output perturbations $\delta\psi$ to the space of linear functions, which, for scalar outputs, is equivalent to the identity function, $\delta\psi(o) = o$. This reduces eq (6) to

$$\forall e \in \mathcal{E}, \quad \delta\mathcal{L}_e(f) = \mathbb{E}_e[\partial_o l(o, y) \cdot o] = 0. \tag{9}$$

The reduced constraints in eq (9) are identical to IRM-s in Kamath et al. (2021).[5] In Arjovsky et al. (2019), this reduced formulation is termed IRM-v1 when the constraints are imposed in a *soft manner*, *i.e.* as squared penalty terms (See eq (16)). In the literature, the term IRM-v1 is widely used for the reduced formulation regardless of whether hard or soft constraints are used, which we adopt here.

However, IRM-v1 has been empirically found to behave quite differently from IRM and fail even in simple problems (Kamath et al., 2021). This failure mechanism can be analytically understood here: Since $\partial_o l(o, y) \cdot o = o(\sigma(o) - y)$ for standard loss functions[4], eq (9) is equivalent to

$$\forall e \in \mathcal{E}, \quad \mathbb{E}_e[oy] = \mathbb{E}_e[o\sigma(o)]. \tag{10}$$

Note that the RHS of eq (10) originates from the RHS of eq (7). Unlike $\sigma(o)$ of eq (7), however, $\mathbb{E}_e[o\sigma(o)]$ is not constant, since it involves expectation that depends on the environment, and therefore it fails to mediate any meaningful invariance relationship. In Supplementary Materials, we generalize this result to the wider class of perturbations that are mixtures of nonlinear basis functions (See Supplementary Materials B).

**Fundamental flaw of IRM**  The above analysis shows that IRM's indirect mechanism for attaining invariance through a shared intermediate term is in fact quite fragile, which easily breaks when the function space of $\delta\psi$ (or $\psi$) gets restricted. We identify this as the *fundamental design flaw of IRM*.

### 3.2 MRI Paradigm

We now introduce a complementary notion of invariance by considering infinitesimal perturbations on label, which we call the *Mirror Reflected IRM*, or *MRI*.

**Definition 3.5.** A predictor $f : X \to O$ is *MRI invariant* over a set of environments $\mathcal{E}$, if the change in risk due to arbitrary infinitesimal label perturbations is *shared* across environments, *i.e.*

$$\forall e_1, e_2 \in \mathcal{E}, \quad \delta\mathcal{L}_{e_1}(f) = \delta\mathcal{L}_{e_2}(f) \tag{11}$$
$$\text{where} \quad \delta\mathcal{L}_e(f) \equiv \lim_{\epsilon \to 0} \mathbb{E}_{P_e(x,y)}[l(o, y + \epsilon\delta\psi(y)) - l(o, y)]/\epsilon$$
$$= \mathbb{E}_{P(y)}[\mathbb{E}_e[\partial_y l(o, y)|y] \cdot \delta\psi(y)] \tag{12}$$

where $\delta\psi : Y \to Y$ is an arbitrary perturbation that is unrestricted in the space of all measurable functions, and $\delta\mathcal{L}_e(f)$ denotes the resulting change in risk. $\mathbb{E}_e[\partial_y l(o, y)|y] \equiv \mathbb{E}_{P_e(o|y)}[\partial_y l(o, y)]$.

**Lemma 3.6.** *For standard loss functions, Definition 3.5 is equivalent to*

$$\forall e_1, e_2 \in \mathcal{E}, \quad \mathbb{E}_{e_1}[o|y] = \mathbb{E}_{e_2}[o|y], \tag{13}$$

*which conserves the label-conditioned feature expectation* $\mathbb{E}_e[o|y] \equiv \mathbb{E}_{P_e(o|y)}[o]$ *across environments.*

*Proof.* Eq (11) is satisfied if and only if $\mathbb{E}_e[\partial_y l(o, y)|y]$ is conserved. For standard loss functions, the loss derivative has the form $\partial_y l(o, y) = -o + \rho(y)$[6]. Therefore, $\mathbb{E}_{e_1}[\partial_y l(o, y)|y] - \mathbb{E}_{e_2}[\partial_y l(o, y)|y] = \mathbb{E}_{e_1}[o|y] - \mathbb{E}_{e_2}[o|y] = 0$. $\square$

**Remark**  Note that MRI attains invariance in a direct manner without involving any intermediate term, which results in the number of constraints in eq (11) matching the conservation law eq (13).

---

[5]IRM-s imposes $\partial_\psi \mathbb{E}_e[l(\psi \cdot o, y)] = 0$, where $\psi$ is a scalar factor. Note that $\partial_\psi l(\psi \cdot o, y) = \partial_o l(o, y) \cdot o$.

[6] Square loss and BCE loss are considered: $\rho(y) = y$ for square loss, and $\rho(y) = 0$ for BCE loss.

| Algorithm | Constraint $\vec{c}(f)$ | $\delta\mathcal{L}_e(f)$ |
|-----------|:-----------------------:|:------------------------:|
| IRM-v1 | $\delta\vec{\mathcal{L}}(f)$ | $\mathbb{E}_e[\,\partial_o l(o,y) \cdot o\,]$ |
| MRI-v1 | $Q\,\delta\vec{\mathcal{L}}(f)$ | $\mathbb{E}_e[\,\partial_y l(o,y) \cdot y\,]$ |

Table 1: Constraint functions of IRM-v1 and MRI-v1. $o = f(x; w)$.

### 3.2.1 MRI-v1

Restricting the label perturbations to the space of linear functions, or equivalently, an identity function $\delta\psi(y) = y$, reduces eq (12) to $\delta\mathcal{L}_e(f) = \mathbb{E}_e[\,\partial_y l(o,y) \cdot y\,]$. This reduces the conservation law eq (13) to

$$\forall e_1, e_2 \in \mathcal{E}, \qquad \mathbb{E}_{e_1}[oy] = \mathbb{E}_{e_2}[oy]. \tag{14}$$

which describes a necessary condition for the MRI invariance eq (13). Therefore, MRI's direct mechanism for attaining invariance continues to hold when $\delta\psi$ is restricted to the space of linear functions.

## 4 Methods

### 4.1 Constrained optimization problem

The full methods of IRM-v1 and MRI-v1 can be formalized as a constrained optimization problem

$$\min_{f \in \mathcal{F}} \mathcal{L}_{\mathrm{tr}}(f) \qquad \text{subject to } \vec{c}(f) = \vec{0}, \tag{15}$$

where $\mathcal{L}_{\mathrm{tr}} = \frac{1}{|\mathcal{E}_{\mathrm{tr}}|} \sum_{e \in \mathcal{E}_{\mathrm{tr}}} \mathcal{L}_e$ is the average risk over the set of training environments $\mathcal{E}_{\mathrm{tr}} \subset \mathcal{E}$. The constraint functions are $\vec{c} = \delta\vec{\mathcal{L}} \in \mathbb{R}^{|\mathcal{E}_{\mathrm{tr}}|}$ for IRM-v1 and $\vec{c} = Q\,\delta\vec{\mathcal{L}} \in \mathbb{R}^{|\mathcal{E}_{\mathrm{tr}}|-1}$ for MRI-v1, where $\delta\vec{\mathcal{L}} \in \mathbb{R}^{|\mathcal{E}_{\mathrm{tr}}|}$ is a vector of perturbed risks $\delta\mathcal{L}_e$ for $e \in \mathcal{E}_{\mathrm{tr}}$, and $Q \in \mathbb{R}^{(|\mathcal{E}_{\mathrm{tr}}|-1) \times |\mathcal{E}_{\mathrm{tr}}|}$ is an orthonormal matrix that satisfies $Q\,\vec{1} = \vec{0}$, such that $Q\,\delta\vec{\mathcal{L}}$ computes the differences of $\delta\mathcal{L}_e$ between environments. For example, for a training set of two environments $\mathcal{E}_{\mathrm{tr}} = \{e_1, e_2\}$, $Q\,\delta\vec{\mathcal{L}} = (\delta\mathcal{L}_{e_1} - \delta\mathcal{L}_{e_2})/\sqrt{2}$, since $Q = [1, -1]/\sqrt{2}$. See Table 1.

### 4.2 Soft-constraint methods

Numerically solving IRM-v1 and MRI-v1 requires converting the hard constraints $\vec{c}(f) = \vec{0}$ to soft constraints, which allows the use of off-the-shelf gradient-based optimization algorithms.

**Penalty Method (PM)** Penalty method is the most commonly used approach, including in Arjovsky et al. (2019), which adds the squared residual constraints as a penalty term to the objective,

$$\min_{f \in \mathcal{F}} \mathcal{L}_{\mathrm{tr}}(f) + \mu \, \|\vec{c}(f)\|^2 \tag{16}$$

However, this method requires increasing $\mu^t \to \infty$ over training iteration $t$ in order to approximate the exact hard-constraint, which leads to training instability and slow convergence (Bertsekas, 1976).

**Augmented Lagrangian Method (ALM)** ALM was introduced to overcome the limitations of penalty method (Bertsekas, 1976), which adds a Lagrange multiplier term to (16)

$$\min_{f \in \mathcal{F}} \mathcal{L}_{\mathrm{tr}}(f) + \mu \, \|\vec{c}(f)\|^2 + \vec{\lambda}^\mathsf{T} \cdot \vec{c}(f), \tag{17}$$

where $\vec{\lambda}$ is typically initialized at $\vec{0}$ and updated at each training iteration $t$ to accumulate the residual constraints $\vec{c}(f(w^t))$. In practice, ALM can operate with moderate values of $\mu$ ($\sim 10$) without fine tuning, and thus exhibits fast and stable convergence (Bertsekas, 1976).

# 5 Analytic Results

## 5.1 General Linear SEM

In this section, we demonstrate that MRI-v1 can effectively eliminate all features that are spuriously correlated with the label in a linear predictor, given a sufficient number of environments. Consider a data generating process according to Fig 1, in which the observation $x = g(z_i, z_s)$ is an injective linear function of the latent features $z_i \in \mathbb{R}^{d_i}, z_s \in \mathbb{R}^{d_s}$. Note that this Structural Equation Model (SEM) (Pearl, 2009) does not require any assumptions on the generation process $Y \to Z_i, Z_s$, which generalizes the SEM of Rosenfeld et al. (2020), which additionally assumed binary labels and additive Gaussian noise for generating the latent features.

We consider a linear predictor $f : X \to O$. Since $g$ is injective and has an inverse over its range, without loss of generality, we can define $f$ as a linear function directly over the latents as

$$o = f(x; w) = w_i^\intercal \cdot z_i + w_s^\intercal \cdot z_s \tag{18}$$

with parameters $w \equiv \{w_i \in \mathbb{R}^{d_i}, w_s \in \mathbb{R}^{d_s}\}$.

**Theorem 5.1.** *Given $|\mathcal{E}_{tr}| > d_s$ training environments, and that $\mathbb{E}_e[z_s y]$ are in general linear positions, MRI-v1 will eliminate all spurious feature dimensions.*

*Proof.* MRI-v1's constraint eq (14) yields

$$\forall e \in \mathcal{E}_{\text{tr}}, \quad \mathbb{E}_e[oy] = w_i \cdot \mathbb{E}_e[z_i y] + w_s \cdot \mathbb{E}_e[z_s y] = const,$$

which can be expressed in a matrix form as

$$w_s \cdot M' = 0, \tag{19}$$

where $M' = M - \bar{M} \cdot 1$ with $M \equiv [\mathbb{E}_e[z_s y]]_{e \in \mathcal{E}_{\text{tr}}} \in \mathbb{R}^{d_s \times |\mathcal{E}_{\text{tr}}|}$ and $\bar{M} = \frac{1}{|\mathcal{E}_{\text{tr}}|} \sum_{e \in \mathcal{E}_{\text{tr}}} \mathbb{E}_e[z_s y]$.

Since $rank(M') = d_s$, eq (19) is equivalent to $w_s = 0$.

$\square$

A similar result was shown for IRM-v1, but under a more restricted setting, such as a specialized linear family of environments with binary labels and additive Gaussian noise (Rosenfeld et al., 2020). In fact, IRM-v1 has been shown to fail in more general linear problems with non-Gaussian noise (Arjovsky et al., 2019; Kamath et al., 2021).

## 5.2 Minimal Example: $d_i = 1$, $d_s = 1$, $|\mathcal{E}_{\text{tr}}| = 2$

Here, we demonstrate a minimal case of the above general linear problem that involves one invariant feature, one spurious feature and two training environments (Shape-Texture linear regression problem in Section 6.1). See Supplementary Materials for the detailed experimental set up and the analytical solutions. A similar minimal examples for linear binary classification are also analyzed and shown in Supplementary Materials (Fig 6, linear shape-texture classification and toy-CMNISTa/b).

**Analytic solutions (Hard-constraints, Fig 2A)** IRM-v1 has two quadratic equality constraints, $\mathbb{E}_{e_1}[o^2 - oy] = \mathbb{E}_{e_2}[o^2 - oy] = 0$, shown as two elliptic curves. The intersection between the non-convex constraints on a 2-D feature space yields a disjoint set of 0-D points, all of which are local constrained optima, including the true invariant optimum, a zero-predictor solution, and two non-invariant solutions. Note that one of the non-invariant solutions exhibits lower train loss than the true invariant optimum solution.

We also test the relaxed version of IRM-v1 by removing the extraneous constraint. The relaxed version has a single constraint $\mathbb{E}_{e_1}[o^2 - oy] = \mathbb{E}_{e_2}[o^2 - oy]$, which describes a pair of hyperbolic curves (appears as two straight lines in Fig 2A), *i.e.* a non-convex constraint. This problem exhibits two local constrained optima, including the true invariant optimum and a non-invariant solution, with the non-invariant solution exhibiting a lower train loss. Therefore, simply relaxing the extraneous constraint does not resolve the fundamental problem of IRM-v1.

In contrast, MRI-v1 has one linear equality constraint, $\mathbb{E}_{e_1}[oy] = \mathbb{E}_{e_2}[oy]$, that exactly prescribes the set of all invariant solutions $w_s = 0$. That is, a solution is an invariant predictor *if and only if it satisfies this constraint.* This is a convex problem, since both the objective and the constraint are convex, and thus features a unique optimum, which is the true invariant optimum solution.

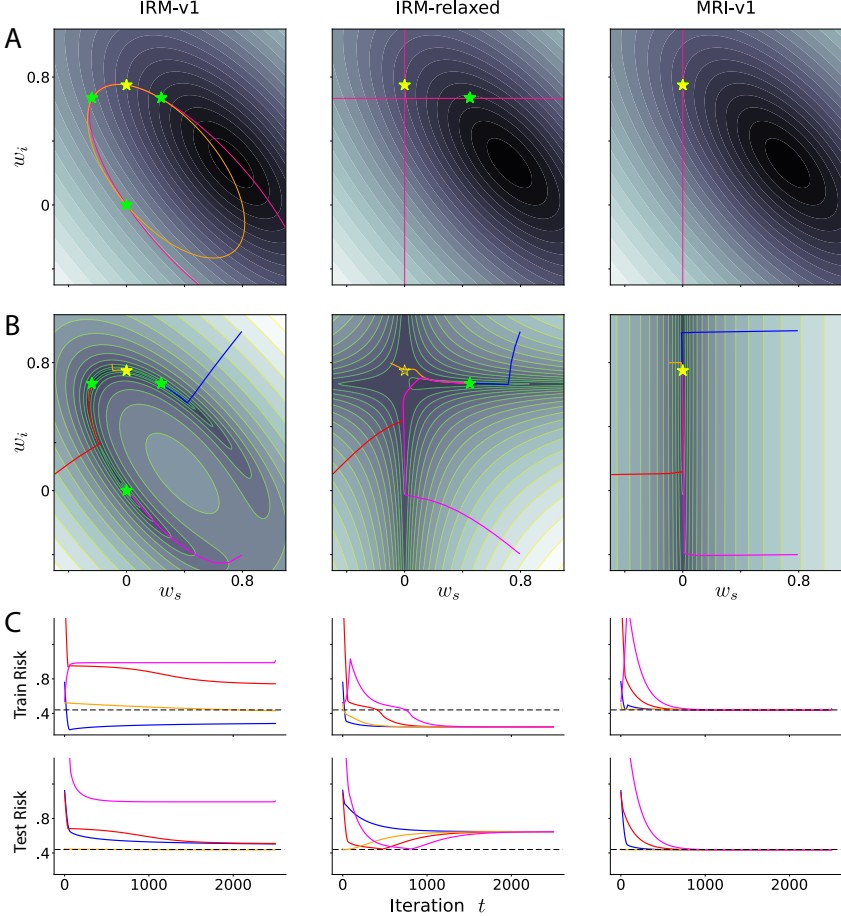

Figure 2: Minimal example in Section 5.2. (A) Hard-constraint case. Background: Average train domain risk $\mathcal{L}_{\text{tr}}(f(w))$. Lines: Invariance constraints $\vec{c}(f(w)) = \vec{0}$. Stars: Local constrained optima. (B) Soft-constraint case. Background: Train loss $\mathcal{L}_{\text{tr}}(f(w)) + \mu \|\vec{c}(f(w))\|^2$. Convergence trajectories $w^t$ from multiple initialization are shown. (C) Loss profiles of the above convergence trajectories. Color-matched with panel B. (Top) Average train domain risk $\mathcal{L}_{\text{tr}}(f(w^t))$. (Bottom) Test domain risk $\mathcal{L}_{\text{test}}(f(w^t))$. Dashed line: True optimal invariant solution (yellow star in panel A/B).

**Convergence Dynamics (Soft-constraints, Fig 2B/C)**    Here, we analyze the optimization dynamics under full-batch gradient descent (penalty method eq (16) with $\mu = 5 \times 10^4$). IRM-v1's convergence is highly dependent on the initialization (shown with differently colored trajectories), due to the presence of multiple local minima. Note that most trajectories do not converge to the true invariant optimum. IRM-relaxed also exhibits complex dynamics due to a saddle point near the true invariant solution: Some trajectories (red and magenta) first approach the line of invariant solutions, but all trajectories eventually converge to the non-invariant solution. Note that the true invariant optimum solution is not even a local optimum of IRM-relaxed at this value of $\mu$, but it would exist in the limit $\mu \to \infty$. In contrast, MRI-v1 always converges to the true invariant optimum regardless of initialization since it is a unique minimum.

# 6    Nonlinear Image-based Problems

Unlike for linear problems, theoretical proof for invariance is difficult to show for nonlinear problems. Here, we empirically investigate the performance of IRM-v1 and MRI-v1 in nonlinear image-based problems.

## 6.1    Datasets

**Shape-Texture Dataset**   We introduce a new dataset that is designed to evaluate domain generalization algorithms across various settings, including linear regression, linear classification, nonlinear image-based regression, and nonlinear image-based classification. The generative process of the dataset involves an invariant feature $z_i = e^{i\theta_i}$, a spurious feature $z_s = e^{i\theta_s}$, and a label feature $e^{i\theta_y}$, each of which represents an orientation on a complex unit circle: $e^{i\theta} \in \mathcal{S}^1$. The angles of orientations are generated as $\theta_y \sim \mathcal{U}_{S^1}$, where $\mathcal{U}_{S^1}$ is the circular uniform distribution, and for $* \in \{i, s\}$, $\theta_* = \theta_y$ with probability $p_*$ or $\theta_* \sim \mathcal{U}_{S^1}$ with probability $1 - p_*$. The parameter $p_i = 0.75$ is fixed across environments, whereas $p_s$ varies from one environment to another. We consider two training environments $\mathcal{E}_{\text{tr}} = \{e_1, e_2\}$ with $p_{s_{e_1}} = 1$, $p_{s_{e_1}} = 0.8$ and one testing environment $\mathcal{E}_{\text{test}} = \{e_0\}$ with $p_{s_{e_0}} = 0$.

In the linear regression task (section 5.2), the observed input is the concatenated latent features $x = [e^{i\theta_i}, e^{i\theta_s}]$ and the label is $y = e^{i\theta_y}$. In the linear classification task, the input is $x = [\sin(\theta_i), \sin(\theta_s)]$ and the label is $y = H(\sin(\theta_y))$, where $H$ is the sign function. In the nonlinear regression/classification tasks, the observed input is the image composed of two planar waves, in which $\theta_i$ is the orientation of the low frequency wave (*i.e.* shape) and $\theta_s$ is that of the high frequency wave (*i.e.* texture), as shown in Fig 5.

**Colored MNIST (CMNIST)**   CMNIST (Arjovsky et al., 2019) is a synthetic dataset derived from MNIST for binary classification. In this dataset, the label $y$ assigned to an image is based on the digit bit $z_i$ (1 for digits $0 \sim 4$ and -1 for $5 \sim 9$) such that $y = z_i$ with probability $p_i$ or $-z_i$ with probability $1 - p_i$. The color bit $z_s$ (1 for red -1 for green) is chosen based on the label such that $z_s = y$ with probability $p_s$ or $-y$ with probability $1 - p_s$. We consider two versions, CMNISTa and CMNISTb, with two sets of environmental parameters: CMNISTa is the version from Arjovsky et al. (2019) with $p_i = 0.75$, $p_{s_{e_1}} = 0.9$, $p_{s_{e_1}} = 0.8$, and $p_{s_{e_0}} = 0.1$ for the training $\mathcal{E}_{\text{tr}} = \{e_1, e_2\}$ and the testing $\mathcal{E}_{\text{test}} = \{e_0\}$ environments; CMNISTb uses $p_i = 0.9$, $p_{s_{e_1}} = 1$, $p_{s_{e_1}} = 0.8$, $p_{s_{e_0}} = 0.1$. In the nonlinear tasks, the input observation $x$ is the colored MNIST image. In the abstracted versions, called toy-CMNISTa/b, the input observation is the two-bit data $x = [z_i, z_s]$ (Kamath et al., 2021).

**Remark**   Note that both Shape-Texture and CMNIST datasets can be equally understood as being generated from Fig 1 with causal directions $Y \to Z_i$ or $Z_i \to Y$ (Fig 9).

## 6.2   Result

Here, we report the performance of IRM-v1, MRI-v1, as well as the vanilla Empirical Risk Minimization (ERM) algorithm (*i.e.* without imposing any invariance constraint). For reference, the results are compared to the *Oracle* performance, which is obtained by applying ERM on the modified training datasets in which the spurious features are rendered uncorrelated with the label.

We tested the algorithms under a wide range of hyperparameters for both the Shape-Texture (Fig 3) and the CMNIST-b (Fig 4) datasets. Overall, MRI-v1 consistently achieves good invariant performance close to the Oracle, whereas IRM-v1 often shows either chance-level performance or poor OOD generalization with large differences between the train and the test domain risks. The results for a specific hyperparameter setting is shown in Table 2.

Under the PM setting, we observe that IRM-v1 often drives the models to the *zero-predictor* solution, *i.e.* making zero output regardless of the input, even with annealing the penalty term to be applied only in the later phase of training. This explains IRM-v1's identically low performance across train and test domains, consistent to the previously reported results in Gulrajani and Lopez-Paz (2020). In contrast, MRI-v1 never drives the models to the zero-predictor solution, consistent with the finding in Sec 5.2 that MRI-v1 does not have an local minima at the zero-predictor in the linear problem setting.

Interestingly, the ALM setting greatly improves IRM-v1's performance especially in terms of accuracy, while MRI-v1's performance remains relatively unchanged between PM and ALM. Under the ALM setting, IRM-v1 shows comparable or slightly higher accuracy than MRI-v1 for the CMNIST dataset.

In the Supplementary Materials, we also report the results for other recent domain generalization methods including MMD (Li et al., 2018b), GroupDRO (Sagawa et al., 2019), and IB-IRM (Ahuja et al., 2021) on the Shape-Texture classification task, which exhibit significantly worse performance than MRI-v1 (Fig 7).

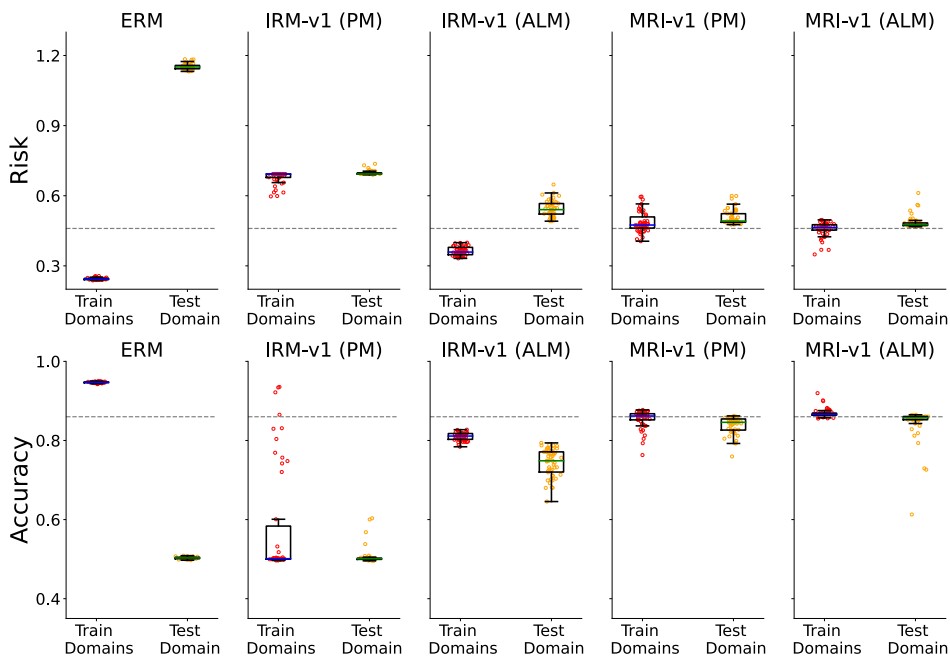

Figure 3: Comparison of algorithms' performance over a range of hyperparameters on the Shape-Texture classification dataset. (Top) averaged train and test domain risk and (bottom) accuracy. The grey dashed line denotes the Oracle performance. Box-plots show sample quartiles.

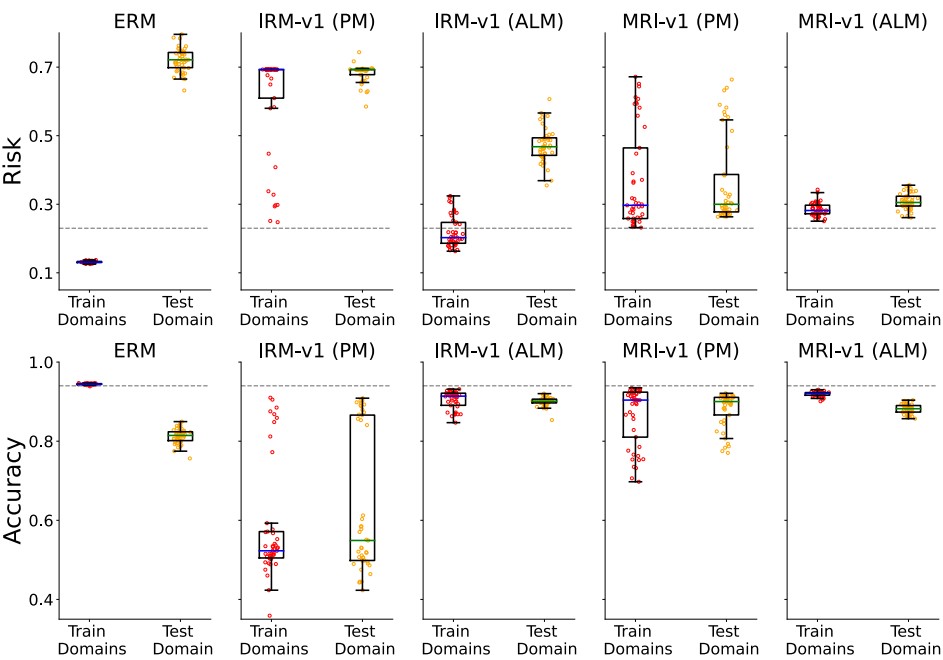

Figure 4: Result on CMNIST-b dataset. Same as Fig 3.

**Risk**

| | S-T Regression | | S-T Classification | | CMNISTa | | CMNISTb | |
|---|---|---|---|---|---|---|---|---|
| | Train | Test | Train | Test | Train | Test | Train | Test |
| Oracle | $0.46 \pm 0.00$ | $0.46 \pm 0.00$ | $0.47 \pm 0.00$ | $0.47 \pm 0.00$ | $0.57 \pm 0.00$ | $0.58 \pm 0.00$ | $0.22 \pm 0.00$ | $0.24 \pm 0.01$ |
| ERM | $0.16 \pm 0.00$ | $1.37 \pm 0.01$ | $0.24 \pm 0.00$ | $1.16 \pm 0.01$ | $0.36 \pm 0.00$ | $1.44 \pm 0.01$ | $0.13 \pm 0.00$ | $0.73 \pm 0.01$ |
| IRM-v1 (PM) | $1.00 \pm 0.00$ | $1.00 \pm 0.00$ | $0.69 \pm 0.00$ | $0.69 \pm 0.00$ | $0.69 \pm 0.00$ | $0.69 \pm 0.00$ | $0.69 \pm 0.00$ | $0.69 \pm 0.00$ |
| IRM-v1 (ALM) | $0.23 \pm 0.00$ | $0.62 \pm 0.01$ | $0.4 \pm 0.01$ | $0.51 \pm 0.01$ | $0.62 \pm 0.02$ | $0.69 \pm 0.02$ | $0.17 \pm 0.01$ | $0.47 \pm 0.02$ |
| MRI-v1 (PM) | $0.53 \pm 0.03$ | $0.54 \pm 0.03$ | $0.44 \pm 0.01$ | $\mathbf{0.46} \pm 0.01$ | $0.62 \pm 0.01$ | $0.66 \pm 0.01$ | $0.46 \pm 0.01$ | $0.4 \pm 0.01$ |
| MRI-v1 (ALM) | $0.45 \pm 0.01$ | $\mathbf{0.46} \pm 0.00$ | $0.47 \pm 0.00$ | $\mathbf{0.47} \pm 0.00$ | $0.63 \pm 0.02$ | $\mathbf{0.64} \pm 0.01$ | $0.25 \pm 0.01$ | $\mathbf{0.29} \pm 0.01$ |

**Accuracy**

| | S-T Classification | | CMNISTa | | CMNISTb | |
|---|---|---|---|---|---|---|
| | Train | Test | Train | Test | Train | Test |
| Oracle | $0.86 \pm 0.00$ | $0.86 \pm 0.00$ | $0.74 \pm 0.00$ | $0.74 \pm 0.00$ | $0.94 \pm 0.00$ | $0.94 \pm 0.00$ |
| ERM | $0.95 \pm 0.00$ | $0.50 \pm 0.00$ | $0.85 \pm 0.00$ | $0.10 \pm 0.00$ | $0.94 \pm 0.00$ | $0.80 \pm 0.02$ |
| IRM-v1 (PM) | $0.52 \pm 0.02$ | $0.52 \pm 0.04$ | $0.73 \pm 0.14$ | $0.23 \pm 0.16$ | $0.6 \pm 0.03$ | $0.52 \pm 0.05$ |
| IRM-v1 (ALM) | $0.79 \pm 0.01$ | $0.77 \pm 0.01$ | $0.64 \pm 0.03$ | $\mathbf{0.66} \pm 0.03$ | $0.93 \pm 0.00$ | $\mathbf{0.91} \pm 0.00$ |
| MRI-v1 (PM) | $0.86 \pm 0.01$ | $\mathbf{0.85} \pm 0.01$ | $0.68 \pm 0.01$ | $0.63 \pm 0.02$ | $0.82 \pm 0.01$ | $0.86 \pm 0.01$ |
| MRI-v1 (ALM) | $0.86 \pm 0.01$ | $\mathbf{0.86} \pm 0.01$ | $0.66 \pm 0.02$ | $\mathbf{0.65} \pm 0.02$ | $0.93 \pm 0.00$ | $\mathbf{0.9} \pm 0.00$ |

Table 2: Comparison of algorithms on the Shape-Texture (S-T) and the Colored MNIST-a/b datasets: (Top) average risk $\mathcal{L}_{\text{tr}}$, $\mathcal{L}_{\text{test}}$, and (bottom) accuracy. Oracle uses environments in which the spurious features are uncorrelated with the label. Mean and standard deviation shown up to 2 decimal places.

## 7 Discussion

**Limitations**   In principle, IRM only requires $\mathbb{E}_e[y|z_i]$ to be constant across domains in order to guarantee invariance, and therefore it has been previously thought to be generally applicable in a wide range of problems, even though this guarantee was only shown in the impractical limit of optimizing over the unrestricted function space of $\psi$. Here, we showed a strong negative result that no meaningful form of invariance can be stated for IRM when the function space of $\psi$ is restricted. In contrast, MRI requires a more limiting condition that the label distribution $P(y)$ and $P(z_i)$ should be constant across domains, but it can be more generally applied even for the case of restricted function space of $\psi$. which yields more practicality.

A common limitation of both MRI and IRM (Rosenfeld et al., 2020; Ahuja et al., 2021) is that they require significant support overlap across domains in order to guarantee OOD generalization. This may limit the applicability of these methods on certain domain generalization benchmarks that consist of domains that lack such overlap, such as VLCS (different image stylization, Fang et al. (2013)), and Terra-Incognita (different natural backgrounds, Beery et al. (2018)). In the Supplementary Materials, we report that both methods do not show significant improvement over ERM on these datasets (Table 3).

**Insensitivity of accuracy metric in evaluating invariance**   For CMINSTa/b, IRM-v1 (ALM) exhibits comparable performance to MRI-v1 in terms of test domain accuracy, despite having significantly worse performance in term of test domain risk (See Table 2). We investigated this phenomenon by analyzing the linear version of the task, toy-CMINSTa/b. The accuracy landscape is piece-wise constant (Fig 8C,F). Especially, it exhibits identical (*i.e.* invariant) accuracy between train and test domain in the region defined by $w_i > |w_s|$. Therefore, the accuracy metric cannot distinguish invariant solutions ($w_s = 0$) from non-invariant solutions within the region. In contrast, the train domain and the test domain risk share the same value only if the solution is invariant ($w_s = 0$) (Fig 8B,E). The constraint function of IRM-v1 (Fig 8A,D) shows that toy-CMINSTa only has invariant local optima and that toy-CMINSTb has additional non-invariant local optima, all of which satisfy $w_i > |w_s|$, thus exhibiting the same accuracy performance as the invariant optimum solution. This result illustrates that using accuracy metric alone for evaluating the degree of invariance could be insufficient, and highlights the need to also consider risk for evaluations.

## Acknowledgments and Disclosure of Funding

This research was done as part of Avinash Baidya's internship at IBM. We thank Joel Dapello for helpful discussions. We also thank the anonymous reviewers for constructive comments.

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
