# OpenReview forum: "The Missing Invariance Principle found --  the Reciprocal Twin of Invariant Risk Minimization "
_NeurIPS.cc/2022/Conference — NeurIPS 2022 Accept_

### Official Review · Reviewer_MsRD · 2022-07-08

**Rating:** 6
**Confidence:** 3
**Soundness:** 3 good
**Presentation:** 2 fair
**Contribution:** 3 good

**Summary:**

This manuscript presents a reciprocal formulation of the Invariant Risk Minimization criterion (Arjovsky et al 2019, arXiv:1907.02893). The authors first present a reformulation of IRM (Def 3.1) as Def 3.2. ("Perturbation-based IRM"), which is invariance  perturbation

They state that Eq.(4) prescribes E-1 constraints, while Eq.(6) (which is equivalent to statements in Def. 3.1 and boldface IRM-v1 defined by Eq.(9) and surrounding text) produces E constraints. The authors claim this extra constraint causes inconsistency in IRM-v1 in comparison with the IRM criterion (eq. 4), which leads to poor performance.

The authors then propose a new invariance condition, MRI (mirror-reflected IRM) which is a perturbation condition on the loss in y (instead of a condition on the output x).

They provide empirical results using an implementation of MRI constraints on a standard architecture (LeNet).

**Questions:**

Hereafter $\mathcal{L} = L$ because mathjax or whatever web framework won't render the subscripts correctly for $\mathcal{L}$...

Questions:

[1] Under the authors perturbation framework, it seems as though MRI is not a mirror of the IRM framework, but simply a different condition. IRM (under perturbation) prescribes a constraint for $\mathcal{L}_e$ under perturbations of prediction output $o=f(x)$ for each $e$. To me, a mirror of this would be a constraint for $L_e$ under perturbations of label $y$ for each $e$. If we assume the analysis after Eq. 6 is correct, clearly this leads to $|\mathcal{E}|$ constraints; as the authors then conclude, this is one too many.

Instead, it seems better to claim this as a corrected or modified criterion. The constraints on $\delta_{y}L_{e_i} = \delta_{y}L_{e_j} $ for $i,j$ pairs in $\mathcal{E}$ produces $\mathcal{E}-1$ effective constraints (due to transitivity). This could similarly be done in $\delta_{o}$, which, to me, would be the natural mirror. Either $\delta_y$ or $\delta_o$ produce the correct number, and, likely the correct formulation on constraints. I think it would be to the benefit of the reader and the authors to rewrite the manuscript with this in mind.

Moreover, from the authors description it is fairly clear (assuming their derivations are correct) that the condition in Eq. 4 and Def 3.1 do not match. Even if meeting Def 3.1 (eq. 6) implies Eq. 4, it is not a necessary condition, shown by constraint cardinality mismatch. Def 3.1 is strictly stronger than Eq. 4. This is not a fault of the authors of this manuscript obviously (it exists in Arjovsky et al 2019) but would aid the author in speaking about the problems of IRM.

[2] Definition 3.2 is a first order rewording of Def 3.1. It thus is only true under certain conditions on $L$. While I think these assumptions are reasonable, they should be explicitly stated in the definition, not just in the last paragraph of Section 2. (Removing these constraints, we could easily imagine an $L$ with some contrived saddlepoint where $\delta L =0$ but the solution sub-optimal.)

[3] Figure 3 suggests that in all but CMNISTb, IRM-relaxed should be performant, but these results do not match Table 1. Are these the same results?

[4] Table 3 should be in the main text? It seems to be a relevant experiment.

Suggestions:

[1] At times the authors leave most of the algebra/context to the reader. At almost every instance of the word "conserved" the authors leave the reader to find out over which variable the expression is conserved. While it eventually becomes discernible given a careful reading in most instances, it would be helpful to be explicit across the board.

[2] Some notation is muddled or inconsistent. $\mathcal{E}$ versus $E$, the usage of $\delta$ in both the test function and as a functional derivative (symbolic only?). The manuscript perhaps should be carefully reconstructed.

[3] Figure 2 makes little sense to me. While it may be my own error, if this is the case with the other reviewers I suggest re-writing/building this figure. I cannot tell what the spurious feature should be, or how the constraint sets are determined. I assume the level-sets are the loss function, but this means it is specific to a single domain.

**Limitations:**

This is a work of theory, and likely does not have direct social impacts outside of machine learning.

**Strengths And Weaknesses:**

Strengths:
* The authors claims appear to be true (and are supported elsewhere in the literature), that IRM is over-specified, and that it can be rephrased into a perturbation method. This rephrasing is, itself, original and interesting from an analytic standpoint, but also it suggests another constraint set, which the authors build into a method (MRI).
* Experimental evidence in the main text backs the authors claims, and improves over IRM.

Weaknesses:
* The paper is couched deep in the perturbation theory literature and notation. While this essentially the authors' main contribution to this particular problem (i.e. the application of perturbations to IRM and the like), many facts and notations go without explanation, making the paper difficult to follow.
* Similarly, the intuition of "mirror reflected" constraint appears to not be true? Instead, this constraint follows a different intuitional path from Arjovsky 2019, one not from Arjovsky Def 3 but from the (un-numbered) pairwise constraint below def 3. This feels confusing and unclear to read.
* The paper shows two experiments, even though the testbed suite they use (DomainBed) provides a large number of experimental conditions. Subsection experiments are relegated to the appendices and somewhat but not entirely match main text results(? It is difficult to tell from plots, but Figure 3 does not appear to match Table 1 numbers.) I am mostly concerned that the strong empirical results will not hold up under other domain shifts, and only work for this form of CMNIST; I would be pleased to be shown wrong.

---

> ### Author Response · Authors · 2022-08-02
> **Addressing your concerns**
>
> Thank you for your review!
>
> > The paper is couched deep in the perturbation theory literature and notation. While this essentially the authors' main contribution to this particular problem (i.e. the application of perturbations to IRM and the like), many facts and notations go without explanation, making the paper difficult to follow.
>
> Thank you for bringing this to our attention. We have revised our paper to explain the notations more clearly.
>
> > Similarly, the intuition of "mirror reflected" constraint appears to not be true? Instead, this constraint follows a different intuitional path from Arjovsky 2019, one not from Arjovsky Def 3 but from the (un-numbered) pairwise constraint below def 3. This feels confusing and unclear to read.
>
> Yes. MRI is indeed mirroring the pairwise constraint in Arjovsky 2019, which we call the conservation law (eq 8 in the revised version), and this conservation law is the essence of IRM’s invariance, which has been shown to be equivalent to the original definition of IRM (Def 3.1) (Kamath et al, 2021).
>
> > The paper shows two experiments, even though the testbed suite they use (DomainBed) provides a large number of experimental conditions. Subsection experiments are relegated to the appendices and somewhat but not entirely match main text results(? It is difficult to tell from plots, but Figure 3 does not appear to match Table 1 numbers.)
>
> Table 2 reports the results of nonlinear, image-based tasks but Figure 3 reports the result of linear abstracted tasks, which are different. Therefore, Table 2 and Figure 3 are not supposed to match.
>
> > I am mostly concerned that the strong empirical results will not hold up under other domain shifts, and only work for this form of CMNIST; I would be pleased to be shown wrong.
>
> Please refer to our common response to the reviewers in which we include additional experiments on VLCS and Terra-Incognita datasets and demonstrate MRI-v1 has the best test performance. In that response, we also discuss the limitations of MRI-v1 for other domain shifts in more detail.

---

> > ### Author Response · Authors · 2022-08-02
> > **Addressing your questions**
> >
> >
> > Questions:
> > > [1]  The constraints on $\delta_y L_{e_i}=\delta_y L_{e_j}$ for i,j pairs in E produces E−1 effective constraints (due to transitivity). This could similarly be done in $\delta_o$, which, to me, would be the natural mirror. Either $\delta_y$ or $\delta_o$ produce the correct number, and, likely the correct formulation on constraints. I think it would be to the benefit of the reader and the authors to rewrite the manuscript with this in mind.
> >
> > We agree with your points. Imposing  $\delta_o L_{e_i}=\delta_o L_{e_j}$, would indeed yield a more corrected version of IRM. We do in fact consider this version of IRM-v1 in section 5.2 of the revised version (called IRM-relaxed). However, we show that this version, while seems promising, does not fully fix the problem of IRM-v1.
> > It yields $\mathbb{E}_{e_i} [oy-o\sigma(o)] = \mathbb{E}_{e_j} [oy-o\sigma(o)]$, which is unrelated to the IRM's result $\mathbb{E}_{e_i}[y|o] = \mathbb{E}_{e_j}[y|o]$.
> >
> > > it seems better to claim this as a corrected or modified criterion.
> >
> > Thank you for the suggestion. We will adapt a better term to reflect this point.
> >
> > > Moreover, from the authors description it is fairly clear (assuming their derivations are correct) that the condition in Eq. 4 and Def 3.1 do not match. Even if meeting Def 3.1 (eq. 6) implies Eq. 4, it is not a necessary condition, shown by constraint cardinality mismatch. Def 3.1 is strictly stronger than Eq. 4. This is not a fault of the authors of this manuscript obviously (it exists in Arjovsky et al 2019) but would aid the author in speaking about the problems of IRM.
> >
> > Yes. This observation is correct, and we indeed argue that this is why IRM-v1 fails. We have re-organized section 3 in the revision, which explains this point more clearly.
> >
> > > [2] Definition 3.2 is a first order rewording of Def 3.1. It thus is only true under certain conditions on L. While I think these assumptions are reasonable, they should be explicitly stated in the definition, not just in the last paragraph of Section 2. (Removing these constraints, we could easily imagine an L with some contrived saddlepoint where δL=0 but the solution sub-optimal.)
> >
> > This is not true. Def 3.2 (now Def 3.3) is indeed equivalent to Def 3.1. The first order condition (i.e. zero gradient) is sufficient, since the loss function is convex. We will explain this for the case of square loss function. ( Note that the same argument applies for BCE loss)
> > The original definition considers a composite predictor $\psi(f(\cdot))$, in which the ‘classifier layer’ $\psi$ maps the feature $f(x)$ to the output. An optimal classifier $\psi: Z \to O$ minimizes the expected loss $E[( y - \psi(z))^2 |f(x)=z]$. By differentiating this loss with respect to $\psi(z)$ and setting the gradient to zero, we obtain $E_e[y|f(x)=z] = \psi(z) = o$ (eq 5 in the revised version). Note that only the first-order condition (i.e. zero gradient) was used in establishing this equivalence. The optimality is automatically guaranteed, since the square loss is convex everywhere.
> > In the variational (perturbation-based) formulation, we consider a simple predictor $f(\cdot)$  and use the first order condition to derive $E_e[y|o] = o$ where $o=f(x)$, which is equivalent to the above result. Does this answer your concern?
> >
> > > [3] Figure 3 suggests that in all but CMNISTb, IRM-relaxed should be performant, but these results do not match Table 1. Are these the same results?
> >
> > As mentioned in the response to your comment in the weaknesses section, Table 2 and Figure 3 demonstrates results for different datasets. Moreover, we do not show IRM-relaxed’s performance in Table 2 (which demonstrates results for non-linear problems) since we already demonstrate the disadvantage of the non-convex constraint provided by IRM-relaxed in the linear problems.
> >
> > > [4] Table 3 should be in the main text? It seems to be a relevant experiment.
> >
> > We couldn’t fit Table 3 in the main text due to the page limit, but it will be included in the camera-ready version. That said, note that, we highlight in our common response to all reviewers that accuracy is not as sensitive enough of a measure as loss is.
> >
> > Suggestions:
> > > [1] At times the authors leave most of the algebra/context to the reader. At almost every instance of the word "conserved" the authors leave the reader to find out over which variable the expression is conserved. While it eventually becomes discernible given a careful reading in most instances, it would be helpful to be explicit across the board.
> >
> > We apologize for this confusion. We have thoroughly updated section 3 for improved clarity.

---

> > > ### Author Response · Authors · 2022-08-02
> > > **Addressing your questions 2**
> > >
> > >
> > > > [2] Some notation is muddled or inconsistent. E versus E, the usage of δ in both the test function and as a functional derivative (symbolic only?). The manuscript perhaps should be carefully reconstructed.
> > >
> > > We have revised through the notations fixed inconsistent ones. $\delta\psi$ and $\delta L_e(f)$ are defined to simply denote an arbitrary perturbation function and the resulting change in risk, respectively, rather than representing functional derivatives. This notation is now further clarified in the revised version.
> > >
> > > > [3] Figure 2 makes little sense to me. While it may be my own error, if this is the case with the other reviewers I suggest re-writing/building this figure.
> > >
> > > We agree Figure 2 was a bit unclear and apologize for the confusion. We have revised section 5.2 and the figure caption for better clarity.
> > >
> > > > I cannot tell what the spurious feature should be, or how the constraint sets are determined. The weight of the spurious feature is shown on the x-axis of Figure 2a/b.
> > >
> > > The constraints of IRM-v1 and MRI-v1 are now more clearly explained in the main text and summarized in Table 1. The constraints of the example task in Figure 2 are shown in eq(25,26,27) of Supplementary Materials.
> > >
> > > > I assume the level-sets are the loss function, but this means it is specific to a single domain.
> > >
> > > The background level-sets show the average risk over the training environments, defined in eq(16) (previously eq(15)).

---

### Official Review · Reviewer_ZSXu · 2022-07-10

**Rating:** 3
**Confidence:** 3
**Soundness:** 1 poor
**Presentation:** 3 good
**Contribution:** 2 fair

**Summary:**

This paper considered the invariant learning for out-of-distribution generation, under the data-generating process with Y \to X. With Fig.1, the invariance regularization is proposed to learn the feature z_i that has invariant relation with y across domains, if the model from latent variable to the input is linear and the number of environments is large enough. Empirical studies are conducted on colored-MNIST and shape-texture linear regression datasets.

**Questions:**

See the weaknesses above.

**Ethics Review Area:**

["I don’t know"]

**Limitations:**

As far as I am concerned, the authors adequately addressed the limitations and potential negative societal impact of their work.


**Strengths And Weaknesses:**

Strengths:
1. The generating process considered in Fig.1 and the followed-up invariance regularization are interesting, which complement the generating process of X \to Y in IRM.
2. This paper is well organized and written.

Weaknesses:
1. The result analysis is MISLEADING. In Table.2, the MRI has a lower loss than IRM and it was claimed that this result is consistent with Table 3. However, in Table 3, the IRM has better accuracy than the proposed MRI method. Indeed, the comparison in terms of accuracy on test domains is more reflectiveness in the considered scenario, as the MRI and IRM optimized different loss functions.
2. This paper highlighted that the proposed method corrected the "flaw" in IRM. However, as far as I'm concerned, this is also misleading, as the two works considered different generating processes. The IRM assumed Y was generated after X (in which the Y is probably given by humans), while the MRI is vice-versa (in which Y is the ground-truth label). In other words, a different definition of Y determines different generating processes,  and it is not necessary for a generation process to be constantly correct. For example, if Y denotes the disease label from the clinicians, then it should be X \to Y. With different generating processes, the invariance and corresponding regularization function can also be different. This can explain the reason for MRI to learn invariant features on the Shape-texture dataset, as the generation of X, and Y follows from Fig.1. In the setting for X \to Y which can also happen, the MRI should not perform better than IRM, in terms of invariant feature selection. Therefore, these two works are not comparable, as the basic underlying causal graph is different.
3. The claim in lines 109-110 that the conservation of Ee[oy]=EPe(o)[Ee[y|o] · o], is a necessary condition for the conservation of Ee[y|o] is wrong. This is because the Pe(o) can also vary across environments, even if Ee[y|o] is invariant, it is not necessarily held that Ee[oy] is invariant.

---

> ### Author Response · Authors · 2022-08-02
> **Addressing your concerns**
>
> > The result analysis is MISLEADING. In Table.2, the MRI has a lower loss than IRM and it was claimed that this result is consistent with Table 3. However, in Table 3, the IRM has better accuracy than the proposed MRI method.
>
> Firstly, we would like to clarify that IRM has comparable accuracy to MRI in Table 2. Secondly, we apologize for the confusion, but we did not claim that the Table 2 results are consistent with that of Table 3. We instead claimed that the accuracy results for the nonlinear dataset in Table 3 are consistent with those of the linear toy versions of the respective datasets such that both IRM and MRI have the same test accuracy, but MRI achieves a significantly lower test loss. Unfortunately, the accuracy results for the linear toy versions were not included. We now include results that shows that IRM and MRI should have the same test accuracy, but different test loss for the linear toy-CMNIST datasets.
>
> > Indeed, the comparison in terms of accuracy on test domains is more reflectiveness in the considered scenario, as the MRI and IRM optimized different loss functions.
>
> Please refer to our common response that discusses why accuracy by itself is not a sensitive enough metric to measure and compare invariance of representations. Additionally, we would like to clarify that in Table 2 we present the average risk for the training environments (train domain loss) and the risk for the test environment ( test domain loss), which doesn't include the penalty term which is the only term that is different between MRI and IRM. Therefore, the loss result presented in Table 2 uses the exact same loss function for ERM, IRM and MRI.
> Given that and the fact that accuracy is not always a sensitive enough metric to measure the invariance of representations, we disagree with your suggestion that accuracy on test domains is more important than loss and we are confident that our results indeed demonstrate the effectiveness of MRI as compared to IRM, in learning invariant representations.
>
> > The IRM assumed Y was generated after X (in which the Y is probably given by humans), while the MRI is vice-versa (in which Y is the ground-truth label). In other words, a different definition of Y determines different generating processes, and it is not necessary for a generation process to be constantly correct. For example, if Y denotes the disease label from the clinicians, then it should be X \to Y. With different generating processes, the invariance and corresponding regularization function can also be different. This can explain the reason for MRI to learn invariant features on the Shape-texture dataset, as the generation of X, and Y follows from Fig.1. In the setting for X \to Y which can also happen, the MRI should not perform better than IRM, in terms of invariant feature selection. Therefore, these two works are not comparable, as the basic underlying causal graph is different.
>
> We would firstly like to clarify that MRI is applicable not only to the causal graph presented in Figure 1 but also to the case in which the invariant node $z_i$ causes the label $y$ ($z_i \to y$). This is the causal graph that Arjovsky et al, 2018 considered on as you mentioned. This detail is now added in the updated Section 2 of our paper. Noticeably, Rosenfeld et al., 2021 considered the $ y \to z_i$ direction for IRM and they also mentioned that either direction $y \to z_i$ or $z_i \to y$ could work. Additionally, both Shape-Texture and CMNISTs datasets can be represented using either directions in the causal graph. Therefore, both IRM and MRI should work on these datasets. The point of this work is to demonstrate that the practical version of IRM, IRM-v1, fails to learn the invariant representation, while MRI-v1 works as expected.
> That said, we do also acknowledge that the domain of applicability of IRM and MRI don’t exactly overlap and we discuss this issue in the common response as well as in our updated limitations section.
>
> > The claim in lines 109-110 that the conservation of Ee[oy]=EPe(o)[Ee[y|o] · o], is a necessary condition for the conservation of Ee[y|o] is wrong. This is because the Pe(o) can also vary across environments, even if Ee[y|o] is invariant, it is not necessarily held that Ee[oy] is invariant.
>
> Thank you for pointing it out. We also noticed this problem after submission and we have now corrected it. In the updated version, it is now explained that
>
> the main problem of IRM is that it uses an indirect way of imposing invariance, i.e. by equaling $ \mathbb{E}_e[y|o]$ to an intermediary term $\sigma(o)$. While this may work for IRM, it does not survive when the output perturbation is restricted to the space of linear functions,
> in which the indirect condition reduces to   $ \mathbb{E}_e[oy] = \mathbb{E}_e[oσ(o)]$ . Since the RHS does not necessarily remain constant across environments, this does not prescribe a meaningful invariance.

---

> > ### Comment · Reviewer_ZSXu · 2022-08-07
> > **Not yet convinced by your rebuttal**
> >
> > Thanks for your feedback. I am still not convinced by your comments regarding the non-reflectiveness of accuracy for out-of-domain generalization and domain of applications in terms of the causal graph. Specifically, as the loss is to optimize towards the accuracy on test domains, the (worst case) test accuracy is the ultimate goal for the OOD problem. If your experiment does show that accuracy currently cannot discriminate the difference between training and test domains, the experiment on more test domains should be conducted. The shape-textured dataset is generated according to Fig.1, therefore it is wrong to say "both Shape-Texture and CMNISTs datasets can be represented using either direction in the causal graph.". More experiments should be conducted to support that in both Z-> Y and Y->Z scenarios, MRI is better than IRM. Otherwise, it is more reasonable to believe that the MRI and IRM are two models, that are respectively more useful for respective scenarios, and hence are not comparable.

---

> > > ### Author Response · Authors · 2022-08-08
> > > **Response to previous comment**
> > >
> > > Thank you for your response.
> > > >This paper highlighted that the proposed method corrected the "flaw" in IRM. However, as far as I'm concerned, this is also misleading, as the two works considered different generating processes.
> > >
> > > We forgot to respond to this comment in the previous response. We do not claim that MRI is a “corrected version of IRM” that is otherwise equivalent. However, we did find the following sentence from abstract that could potentially be misunderstood as such:
> > > - Here, we identify a fundamental flaw of IRM formulation that causes the failure. We then introduce a complementary notion of invariance, MRI, that is based on conserving the class-conditioned feature expectation Ee[f(x)|y] across environments, **that corrects for the flaw in IRM**.
> > >
> > > This has been changed to
> > > - Here, we identify a fundamental flaw of IRM formulation that causes the failure. We then introduce a complementary notion of invariance, MRI, based on conserving the label-conditioned feature expectation Ee[f (x)|y] across environments, **which is free of this flaw**.
> > >
> > > Other than this minor dent, however, it is never claimed nor implied that MRI is a “corrected version of IRM” that is otherwise equivalent. We would appreciate it if you could point out if there are any other expressions that could potentially lead to a misleading impression.

---

> > > ### Author Response · Authors · 2022-08-08
> > > **Further clarification.**
> > >
> > >
> > > >More experiments should be conducted to support that in both Z-> Y and Y->Z scenarios, MRI is better than IRM. Otherwise, it is more reasonable to believe that the MRI and IRM are two models, that are respectively more useful for respective scenarios, and hence are not comparable.
> > >
> > > Our overall impression is that you seems to have somehow misunderstood the main point of the paper. We do not claim that MRI is a “better version of IRM” in every comparable way. The main point of our work is to identify the critical failure mechanism of IRM and demonstrate its implications. MRI is introduced as an alternative principle that is free of this failure mode.
> > >
> > > > The IRM assumed Y was generated after X, while the MRI is vice-versa. .....  This can explain the reason for MRI to learn invariant features on the Shape-texture dataset, as the generation of X, and Y follows from Fig.1. In the setting for X \to Y which can also happen, the MRI should not perform better than IRM, in terms of invariant feature selection. Therefore, these two works are not comparable, as the basic underlying causal graph is different.
> > >
> > > Also, you seem to be proposing that IRM is only meant to work for the $z_i \to y$ case but we use $y \to z_i$ for which IRM is not designed to yield invariance, which is why MRI works better than IRM in our results. This is not true for multiple reasons.
> > >
> > > 1. IRM is supposed to work for both  $z_i \to y$ and $y \to z_i$.
> > >
> > > 2. The main failure mechanism of IRM/IRM-v1 has nothing to do with the causal direction.
> > >
> > > 3. The dataset we use (Shape-texture and CMNIST) can equivalently be generated by both causal directions.
> > >
> > > Therefore, even though IRM and MRI do not have identical range of applicability, this difference is irrelevant to our analysis and it  does not invalidate our result in any way. We expand this below.
> > >
> > > 1. The main failure mechanism of IRM/IRM-v1, which we focus on in this work, is due to the indirect manner IRM imposes invariance: this indirect, over-constrained approach may work for IRM which considers the unrestricted output perturbation function space, but it fails for IRM-v1 which restricts the function space to be linear. Note that this result eq(7~10) is based on eq(2) which applies to both $y \to z_i$ and $z_i \to y$ cases. Therefore, this *failure mechanism has nothing to do with the causal direction between $y$ and $z_i$*.
> > >
> > > 2. With MRI/MRI-v1, we show that a direct approach of imposing invariance does not suffer from the above failure mode when the function space is restricted. This result eq(13~14) is based on eq(3), which only requires the marginal distribution $P(y)$ being constant across environments.
> > >
> > > 3. Shape-Texture dataset can equivalently be generated by either causal directions, since the following two processes yield identical distribution over the label and the latent angles:
> > > - ($Y \to Z_i$) $\theta_y \sim \text{Uniform}(0, 2\pi)$ and $\theta_i=\theta_y$ with probability $p_i$ or $\theta_i \sim \text{Uniform}(0, 2\pi)$ otherwise
> > > - ($Z_i \to Y$) $\theta_i \sim \text{Uniform}(0, 2\pi)$ and $\theta_y = \theta_i$  with probability $p_i$ or $\theta_y \sim \text{Uniform}(0, 2\pi)$ otherwise
> > >
> > > (Similarly, CMNIST can equivalently be generated by either causal directions.)
> > > This equivalence is now shown in Figure 9 (updated version), which compares the joint distributions of shape and label angles that are generated by both processes. Note that the marginal distribution $P(y)$ is constant across environments. These are appropriate settings for both IRM and MRI, in which they can be compared in a fair manner.
> > >
> > > 4. We consider general linear problems (with constant $P(y)$ across environments) and analytically prove that MRI-v1 can always solve these problems (under minor assumptions). We demonstrate this result with a minimal example in which IRM-v1 is shown to fail. Figure 2 illustrates that MRI-v1’s constraint exactly prescribes the set of all invariant predictors, yielding the true constrained optimum, whereas IRM-v1’s constraints induce multiple optima that are inconsistent to any notion of invariance. This comparison clearly demonstrates the failure mechanism of IRM/IRM-v1 and their implications.
> > >
> > > 5. For nonlinear OOD classification tasks, despite the lack of theoretical guarantee, MRI-v1 is shown to exhibit remarkably improved performance over IRM (in terms of test-domain risk, but not always in terms of test-domain accuracy).
> > >
> > > We believe our paper delivers the above results clearly without any misleading information.

---

### Official Review · Reviewer_fDdb · 2022-07-11

**Rating:** 8
**Confidence:** 4
**Soundness:** 3 good
**Presentation:** 4 excellent
**Contribution:** 3 good

**Summary:**

The authors build upon the observations in Kamath et al., 2021 regarding conditional output invariance across environments about Arjovsky's invariant risk minimization (IRM) principle. They derive a novel principle based upon label perturbation equivariance, dubbed Mirror Reflected IRM (MRI).
They prove their new approach can eliminate spurious feature dimensions for the case of a linear structural equation model more general than the one considered in Rosenfeld, et al 2020 regarding IRM-v1. In the nonlinear case, they propose optimization using the Augmented Lagrangian method with constraints on all environment pairs.
On a newly introduced shape-texture dataset and two versions of the colored MNIST dataset introduced in Arjovsky et al 2019, they demonstrate that MRI-v1 outperforms ERM and IRM-v1, while coming close to the performance of an oracle.

**Questions:**

1. Should $\sigma (\psi (o))$ in eq (6) simply be  $\sigma (o)$ ?
2. Do the colors for the plots in Fig. 2 refer to different parameters in the model or to different initial conditions ?
3. Please also note the questions inherent in Weaknesses 2 and 4 above.
4. The authors mention requiring $E$-1 independent pair-wise constraints in Sec. 3.2 and Sec 3.3. It appears that all experiments use $E=2$. Does $E>2$ increase the computational burden of MRIv1 versus IRM ? Have the authors does any experiments with $E>2$ ?


**Limitations:**

The authors point out the limitations of their approach in Sec. 6.
It is not clear if question 3 (weakness 2 and 4) mentioned above indicates other limitations.

**Strengths And Weaknesses:**

1. The novel approach is clearly presented and motivated quite well. The theoretical results appear to be sound, although I have not checked the proofs in the appendix.

2. The empirical results confirm the advantages of MRI-v1 over IRM-v1.

Weaknesses

1. The empirical improvement of MRI-v1 over IRM-v1 in the linear regime in Table 2 appears to be quite modest.
The authors might be able to provide more convincing experiments perhaps with a dataset where IRM-v1 fails more radically, or in the non-linear regime or with architectures more complex than LeNet.

2. Results such as Lemma 3.5 seem to rely on either the binary cross-entropy loss or the squared loss and it is not clear if the author's approach is applicable to other loss functions

3.
The authors call their approach mirror reflection because they consider label perturbations instead of output perturbations.
However instead of the same loss across environments, they require the same change in loss across environments.
That reminds this reviewer of equivariance instead of invariance as defined in standard statistical texts such as Casella and Berger.
So I have dubbed their approach MRE instead of MRI.

4. Minor clarity issue:
It is not clear how the penalty parameter $\lambda$ or the ALM parameter $\mu$ in the experiments was tuned or how difficult it was to tune these parameters.

5. Typo: Heavyside should be Heaviside in Sec. 5.2.1.

6. SEM is not defined prior to its use in Sec. 4.1

---

> ### Author Response · Authors · 2022-08-02
> **Addressing your concerns**
>
> Thank you for your review!
>
> > Empirical improvement of MRI-v1 over IRM-v1 in the linear regime in Table 2 appears to be quite modest. Try other dataset/non-linear regimes/neural architectures.
>
> We include results on additional experiments on VLCS and Terra Incognita datasets in the supplementary section. Please refer to the common response above in which we also discuss why the results Table 2 are not modest.
>
> > Loss-dependency of the Lemma/result.
>
> Yes, Indeed the proof of Lemma 3.2 of the original IRM formulation (Kamath et al 2020) and the proof of Lemma 3.4 of the perturbation-based IRM are based on the property of standard convex loss functions (including square loss and BCE loss)
> that the optimal $\psi$ of the risk $E[ l(\psi(z),y) | f(x)=z]$ satisfies $E[y| f(x)=z] = \sigma(\psi(z))$,
> which is based on the fact that the derivative $\partial_o l(o,y)$ has the form $-y + \sigma(o)$.
> This is indeed not a general property satisfied by all loss functions, which is another limitation of IRM.
>
>
> > MRI requires same change in loss across environments. This is equivariance instead of invariance.  The authors call their approach mirror reflection because they consider label perturbations instead of output perturbations. However, instead of the same loss across environments, they require the same change in loss across environments. That reminds this reviewer of equivariance instead of invariance as defined in standard statistical texts such as Casella and Berger. So I have dubbed their approach MRE instead of MRI.
>
> By invariance, we were referring to conservation law of E[o|y] across domains.
>
> > Tuning of ALM parameters.
>
> ALM parameters didn’t require extensive tuning, especially for MRI, as demonstrated in Figure 7 (previously Figure 6) of the updated version.
>
> > Typo: Heavyside should be Heaviside in Sec. 5.2.1.
>
> Corrected.
>
> > SEM is not defined prior to its use in Sec. 4.1
>
> Corrected.
>
> Questions:
> > Should σ(ψ(o)) in eq (6) simply be σ(o)?
>
> Yes, we’ve corrected this in the revision.
>
> > Do the colors for the plots in Fig. 2 refer to different parameters in the model or to different initial conditions?
>
> It is for different initial conditions. We added this clarification in the figure caption.
>
> > Please also note the questions inherent in Weaknesses 2 and 4 above.
>
> Please refer to our response to your earlier comments.
>
> > The authors mention requiring E-1 independent pair-wise constraints in Sec. 3.2 and Sec 3.3. It appears that all experiments use E=2. Does E>2 increase the computational burden of MRIv1 versus IRM ? Have the authors done any experiments with E>2?
>
> The computational cost of MRI-v1 grows linearly as |E|-1, due to transitivity. We have indeed ran experiments with |E|>2, but they weren’t needed for this paper.

---

> > ### Comment · Reviewer_fDdb · 2022-08-09
> > **Satisfactory response from the authors**
> >
> >
> > The authors have adequately addressed my concerns.
> > The experimental comparison in Tables 4 and 5 for two additional DomainBed datasets lends additional credibility to the claim on test loss reduction (but not necessarily accuracy).
> > I am also glad they have added comparisons of the Augmented Lagrangian Method (ALM) with the Penalty Method for both IRM-v1 and MRI-v1 to illustrate the improvements in accuracy by switching to ALM as well as comparisons with other methods such as MMD and GroupDRO..
> > My rating remains unchanged by the discussion between authors and reviewers thus far.

---

### Official Review · Reviewer_ZMnn · 2022-07-13

**Rating:** 7
**Confidence:** 3
**Soundness:** 3 good
**Presentation:** 3 good
**Contribution:** 3 good

**Summary:**

The paper shows that IRM enforces one more constraint than necessary from viewpoint of the invariance principle, resulting in its failures. Then, a modified paradigm is presented along with a practical version which optimizes the correct set of constraints. Theoretical guarantees are provided in the linear setting and experiments on simple tasks show benefits and the optimization properties of the approach.

**Questions:**

1. First line of Equation (9) is missing a square bracket. But I am also confused as to why the output perturbations should be linear and not constant when assuming linear functions $\psi$.

2. Equation (15) or Table 1 3rd column should explicitly specify what $\vec{c}$ is. It is also better to provide more details on how to estimate the relevant quantities in the constraint.

3. Was IRMv1 implemented using the constraints in equation (15) by estimating the perturbed risks or using the original implementation?

4. Line 142: I am not clear on what the assumption on noise is here. Is it still additive but need not be Gaussian?
5. Equation (20) has an undefined $\vec{c} (\theta)$ term. Should it be $\vec{c}(f(\theta))$?

6. The “Convexity matters” paragraph seems to claim about convexity more generally, which I believe is not backed by the experiments which were solely done in the specific context of IRM-type losses. I think the claims should be appropriately rewritten.

**AFTER REBUTTAL**: Authors have addressed my concerns and I have improved my score.

**Limitations:**

Yes.

**Strengths And Weaknesses:**

Strengths:

1. Paper is clearly written and well motivated, showing the precise reason for IRM’s failure in certain scenarios.
2. The equivalent perturbation-based formulation can be helpful for easier analysis of IRM-type methods.
3. The qualitative experiments on Shape-Texture dataset in section 4.2 are illuminating; they show how IRM’s objective can induce multiple optima that may use the spurious features.

Weaknesses:

1. While the paper has a good set of qualitative experiments on synthetic tasks, it can be improved with more realistic experiments, for example, on the domain generalization benchmark DomainBed [1]. This will also help in clearly identifying if the performance of MRI is better than IRM; currently, the test classification accuracies are on par with IRM for most tasks (Table 3).
2. It would also be good to see how the proposed method fares to other more recent domain generalization methods  (e.g., GroupDRO [2], MMD[3], IB-IRM [4], etc.).
3. Practical implementation of the proposed method is not fully specified; for example, clear definition of constraints $\vec{c}$ for MRI, how perturbed risks are estimated empirically, etc. Additionally, it seems that IRMv1 is implemented with this reformulation (using constraints) rather than the original implementation, and may not be a fair comparison.

---

> ### Author Response · Authors · 2022-08-02
> **Addressing your concerns**
>
> Thank you for your review!
>
> > While the paper has a good set of qualitative experiments on synthetic tasks, it can be improved with more realistic experiments, for example, on the domain generalization benchmark DomainBed [1]. This will also help in clearly identifying if the performance of MRI is better than IRM; currently, the test classification accuracies are on par with IRM for most tasks (Table 3).
>
> We now perform additional experiments on VLCS and Terra Incognita datasets and include them in the supplementary materials. Please refer to our common response, where we also discuss how accuracy by itself can be a misleading measure for comparing different domain generalization algorithms.
>
> > It would also be good to see how the proposed method fares to other more recent domain generalization methods (e.g., GroupDRO [2], MMD[3], IB-IRM [4], etc.).
>
> We now include a comparison between MRI and these methods (GroupDRO, MMD, IB-IRM) on the Shape-Texture dataset in Figure 5 of the updated version. We demonstrate that MRI performs better than all these methods. This matches our expectation based on the previous result by Gulrajani et al., 2020 in which they demonstrated that these other methods provide no improvement over ERM in Gulrajani et al., 2020 on the CMNIST dataset.
>
> > Practical implementation of the proposed method is not fully specified; for example, clear definition of constraints  $\vec{c}$ for MRI, how perturbed risks are estimated empirically, etc. (Q: Equation (15) or Table 1 3rd column should explicitly specify what $\vec{c}$ is. It is also better to provide more details on how to estimate the relevant quantities in the constraint.)
>
> This is now clarified in the revised version. The constraints of IRM/MRI-v1 are computed by computing the following
>
> IRM-v1: $\delta L_e =  \mathbb{E}_e [ o \cdot \partial l/\partial o] $, constraint: $\vec{c} =  \vec{\delta L}$
>
> MRI-v1: $\delta L_e =  \mathbb{E}_e [ y \cdot \partial l/\partial y]  $, constraint: $\vec{c} =  Q\vec{\delta L}$
>
> where $Q \in \mathbb{R} ^{(|\mathcal{E}_\text{tr}|-1) \times |\mathcal{E}_\text{tr}|}$
> is an orthonormal matrix that satisfies $Q \, \vec{1} = \vec{0}$.
>
> > Additionally, it seems that IRMv1 is implemented with this reformulation (using constraints) rather than the original implementation, and may not be a fair comparison. (Q: Was IRMv1 implemented using the constraints in equation (15) by estimating the perturbed risks or using the original implementation?)
>
> The reformulated IRM-v1 is identical to the original version. We clarify this in the revised version.
>
> Questions:
> > First line of Equation (9) is missing a square bracket. But I am also confused as to why the output perturbations should be linear and not constant when assuming linear functions ψ.
>
> Perturbations of linear functions are also linear functions:
> e.g.   $\psi_1(y) = a y$, $\psi_2(y) = by$ $\to  \psi_1(y) - \psi_2(y) = (a-b)y$.
>
> > Line 142: I am not clear on what the assumption on noise is here. Is it still additive but need not be Gaussian?
>
> No assumption on the noise-generating process is needed for the proof. We clarify this in our revised version.
>
> > Equation (20) has an undefined $\vec{c(θ)}$ term. Should it be  $\vec{c(f(θ))}$?
>
> We have corrected it in the revised version.
>
> > The “Convexity matters” paragraph seems to claim about convexity more generally, which I believe is not backed by the experiments which were solely done in the specific context of IRM-type losses. I think the claims should be appropriately rewritten.
>
> You are right that more experiments and detailed theoretical arguments could be included to demonstrate our claim. Since this is not the main point of our paper, we decided to remove this paragraph from the discussion section.

---

> > ### Comment · Reviewer_ZMnn · 2022-08-08
> > **Thanks for the response**
> >
> > Thanks for your clarifications and additional experiments. I will update my score soon (currently I am having trouble accessing the revised paper).

---

> > ### Comment · Reviewer_ZMnn · 2022-08-09
> > **Improved score + Minor comments**
> >
> > I have improved my score but I have the following minor comments.
> > 1. It would be helpful to have test accuracies reported in Table 2 (instead of Appendix). Even if it is a coarse metric for measuring invariance, it is still the ultimate goal.
> > 2. I am not sure why the authors converted VLCS and TI to binary classification. Is this a limitation of the approach or due to time constraint in rebuttal phase? I think the final version should include results on full VLCS/TI task. Also, since there seems to be space left, these results can be moved to the main paper.

---

### Author Response · Authors · 2022-08-02
**Common response to all reviewers**

We thank the reviewers for their positive feedback on our work. All reviewers found our work to be well presented and written. R1, R2 and R4 found our work to be well-motivated. R1 found our perturbation-based formulation to be helpful for analysis of IRM-type method and R4 found it original and interesting. R1 commented that we show the precise reason for IRM’s failure. R2 and R4 found that the empirical results backed our claims and show improvement over IRM.

We also thank the reviewers for their insightful remarks that have helped improve our manuscript. Below, we will first focus on some of the common issues raised by most of the reviewers and then address each reviewer individually.

> Similar accuracy between MRI-v1 and IRM-v1 in CMNISTa/b

Firstly, we would like to mention that the accuracy for MRI-v1 and IRM-v1 are only similar when they are trained using the ALM method. The original implementation of IRM-v1 was using the penalty method (PM), which is known to require extensive hyperparameter tuning (Gulrajani et al., 2020). In fact, in our simulations, IRM-v1 always converged to the zero-predictor for non-linear datasets for a wide range of penalty coefficient values. We now show the results using PM non-linear datasets in the updated version of Table 2-3 and Figure 5-6.
Using PM, MRI-v1 achieves a far better loss and accuracy than IRM-v1. This difference is due to the fact that zero-predictor is a local constrained optimum for IRM-v1, but it is not for MRI-v1. We originally designed the ALM method in order to improve the convergence dynamics of IRM-v1. In contrast, ALM only marginally improves MRI-v1’s performance compared to PM.

Secondly, we want to point out that often accuracy is not sensitive enough to reliably measure the degree of invariance of a model. For example, consider the linear classification problem of toy-CMINSTb shown in Figure 8. For this problem, MRI-v1 has a unique optimum, which is the optimal invariant solution, while IRM-v1 exhibits multiple local optima, including non-invariant solutions. Yet, all these solutions exhibit identical accuracy (except for the zero-predictor solution of IRM-v1), since the accuracy landscape (Figure 8f) is flat over a wide range of solutions and remains invariant across domains: invariant accuracy is achieved as long as $w_i>abs(w_s)$. This shows that accuracy cannot be used as a measure of invariance in this problem. Therefore, the fact that MRI-v1 and IRM-v1 exhibit similar test-domain accuracy does not necessarily indicate that they achieve a similar degree of invariance. We now include this discussion in the updated version of our paper.


> Limited experiments/More realistic experiments

Firstly, we hope that we have now convinced you that we clearly demonstrate the advantages of MRI-v1 over IRM-v1 in learning invariant representations for both shape-texture and CMNIST datasets. Nevertheless, we are happy to test MRI on more datasets. To that end, we now test our algorithm on two additional datasets, VLCS and Terra Incognita (TI), that are part of DomainBed. We now include these results in the updated version of our manuscript in Tables 4 and 5. For simplicity, we restricted the problem to a binary classification task by choosing two classes for each dataset, namely opossum and raccoon for TI and bird and person for VLCS. In these datasets, we also find that both IRM-v1 and MRI-v1 achieve similar test dataset performance in terms of accuracy, but MRI-v1 achieves a significantly lower test loss than both IRM-v1 and ERM. In fact, IRM-v1 achieves a worse test loss than ERM for both datasets, a behavior that was also observed by Kamath et al., 2021.  Given that MRI achieves the best test dataset loss while achieving similar test accuracy, we can confidently say that MRI performs the best on both datasets.

---

> ### Author Response · Authors · 2022-08-02
> **continued**
>
>
> > Limitation: Domain of applicability
>
> - Common between MRI and IRM: A limitation of both MRI and IRM (Rosenfeld et al., 2020, Ahuja et al., 2021) is that it requires significant overlap in the support of different domain distributions (we previously mentioned this as a limitation of our method). This may limit the applicability of both MRI and IRM on many DomainBed benchmarks. Rotated-MNIST, for example, expects an algorithm to generalize to an orientation of the digit that was never seen in the training set, in which the support of the training and testing environments clearly do not overlap. Similar arguments can be made for other datasets. For example, images with different styles (photos vs sketches) may not satisfy the significant support overlap condition.
>
> - Different between MRI and IRM: IRM and MRI's domain of applicability only coincide when there is no label shift or shift in invariant feature distribution. Synthetic datasets can be implemented that do not follow these conditions such that MRI, but not IRM, would fail. Regardless of the applicability of IRM, however, IRM-v1 is not guaranteed to work with any domain shifts (Ahuja et al., 2020, 2021, Kamath et al., 2020), as we analytically show in our work.
>
> We have updated our limitations section to include these limitations.

---

### Meta-Review · Area_Chair_tHrT · 2022-08-27

**Recommendation:** Accept
**Confidence:** Less certain

**Metareview:**

The paper theoretically considers the formulation of IRM for OOD generalization, and proposes a new MRI as a fix to the flaw of the IRM.  The proposed method is empirically shown to give better test loss.
The evaluations of the reviewers split largely and did not converge, unfortunately.  The paper provides an interesting theoretical analysis of IRM and comparison with MRI, which has a good contribution to the field of invariant learning.  However, some concerns have also been raised.  There may be some overclaim on the relation between IRM and MRI; the underlying assumptions are not identical and the pros and cons of the two formulations should be compared with care.   The evaluation with the test accuracy and test loss should be demonstrated clearly.  These concerns should be addressed carefully in the paper.



**Award:**

No

---

### Decision · Program_Chairs · 2022-09-14

Accept